# Ovarian Cancer—Insights into Platinum Resistance and Overcoming It

**DOI:** 10.3390/medicina59030544

**Published:** 2023-03-10

**Authors:** Andrei Havasi, Simona Sorana Cainap, Ana Teodora Havasi, Calin Cainap

**Affiliations:** 1Department of Oncology, “Iuliu Hatieganu” University of Medicine and Pharmacy, 400012 Cluj-Napoca, Romania; 2Pediatric Clinic No. 2, Department of Pediatric CNardiology, Emergency County Hospital for Children, 400177 Cluj-Napoca, Romania; 3Department of Mother and Child, “Iuliu Hatieganu” University of Medicine and Pharmacy, 400013 Cluj-Napoca, Romania; 4Department of Gastroenterology, County Hospital Oradea, 410169 Oradea, Romania; 5Department of Medical Oncology, The Oncology Institute “Prof. Dr. Ion Chiricuta”, 400015 Cluj-Napoca, Romania

**Keywords:** ovarian cancer, platinum resistance, platinum resistance mechanisms, overcoming platinum resistance

## Abstract

Ovarian cancer is the most lethal gynecologic malignancy. Platinum-based chemotherapy is the backbone of treatment for ovarian cancer, and although the majority of patients initially have a platinum-sensitive disease, through multiple recurrences, they will acquire resistance. Platinum-resistant recurrent ovarian cancer has a poor prognosis and few treatment options with limited efficacy. Resistance to platinum compounds is a complex process involving multiple mechanisms pertaining not only to the tumoral cell but also to the tumoral microenvironment. In this review, we discuss the molecular mechanism involved in ovarian cancer cells’ resistance to platinum-based chemotherapy, focusing on the alteration of drug influx and efflux pathways, DNA repair, the dysregulation of epigenetic modulation, and the involvement of the tumoral microenvironment in the acquisition of the platinum-resistant phenotype. Furthermore, we review promising alternative treatment approaches that may improve these patients’ poor prognosis, discussing current strategies, novel combinations, and therapeutic agents.

## 1. Introduction

Ovarian cancer is a significant cause of morbidity and mortality worldwide, responsible for 300,000 new cases each year and almost as many deaths [1]. Often diagnosed in an advanced stage, it is the most lethal gynecological cancer, with a 5-year survival rate of 26–42%, depending on the initial stage [2]; however, more than 40% of stage III/IV patients die within the first year and 25% within the first 90 days following diagnosis [3]. Ovarian tumors may arise from epithelial, stromal, or germ cells, where over 90% of malignant ovarian tumors arise from epithelial cells [4]. A heterogeneous disease, epithelial ovarian cancer (EOC) comprises several histological subtypes: high-grade serous ovarian cancer (HGSOC) (70–80%), endometrioid (10%), clear cell (10%), mucinous (3%), and low-grade serous (<5%) [5].

Most high-grade serous tumors are sporadic; however, up to 15% of patients with ovarian cancer have a genetic predisposition. BRCA1 and BRCA2 mutations are responsible for most hereditary epithelial ovarian cancer, while Lynch syndrome is associated with clear-cell and endometrioid tumors [6]. Patients with a hereditary predisposition have a younger age at presentation, a history of other cancers, and positive family history. Traditionally patients that fit this profile are considered for genetic testing; however, up to 44% of BRCA carriers have no family history, and the highest annual risk for ovarian cancers for BRCA carriers is for those between 50 and 69 years old [7,8]. Almost all BRCA-mutated cancers are high-grade serous ovarian cancers [9]. However, high-grade serous tumors are not limited to the BRCA mutation. TP53 mutations are present in almost 97% of tumors, and other homologous recombination repair mutations, including EMSY, RAD51, ATM, BARD1, BRIP1, ATR, PALB2, RB1, and CDKN2A, have been identified in 11% of patients [10].

Surgery with complete cytoreduction is the standard of care in the management of EOC. In patients in which resection with no gross residual disease can be achieved, surgery can be offered upfront, followed by adjuvant systemic therapy. For patients with stage III and IV, bulky, extensive disease, neoadjuvant chemotherapy followed by interval debulking surgery and adjuvant chemotherapy was associated with higher R0 resection rates and better survival outcomes than primary debulking surgery [11]. Surgery continues to play an essential role in the management of EOC, even in recurrent disease. Secondary debulking surgery in platinum-sensitive recurrent EOC has been linked to a 5-month progression-free survival (PFS) improvement. This survival benefit was more significant in patients where R0 resection had been achieved [12].

Systemic treatment is essential in the management of ovarian cancer. The EORTC-ACTION [13] and ICON1 [14] trials established almost 20 years ago the superior outcomes associated with the use of platinum and taxane-based chemotherapy in ovarian cancer compared with monotherapy and observation alone. Patient selection for adjuvant chemotherapy depends on histological subtype and tumor grade. In very early-stage IA tumors, observation is recommended for low-grade serous, grade 1,2 endometrioid, and grade 1,2 mucinous ovarian cancer. Nevertheless, there remains a question regarding the benefit of adjuvant treatment for clear-cell carcinoma stage I A, B, and C1; stage IB and IC; grade 1,2 endometrioid; and stage IB/C low-grade serous [15]. Regarding the number of chemotherapy administrations, the standard number of cycles remains six; however, the GOG157 trial reported similar outcomes when using three or six of paclitaxel and carboplatin [16].

The combination of paclitaxel and carboplatin is the standard of care treatment for patients with advanced disease [17]. Other various dose-dense administration schedules and the association of intraperitoneal chemotherapy have been investigated in numerous trials but with conflicting results [18]. Despite adjuvant treatment and optional surgery, patients with advanced ovarian cancer often present high recurrence rates. Therefore, several trials have investigated the use of maintenance therapies; paclitaxel marginally improved progression-free survival (PFS) [19], and maintenance with bevacizumab proved to provide benefits only in PFS terms, except for a subgroup of poor-prognosis patients where a trend toward improved overall survival (OS) could be observed [20]. Additionally, for BRCA-mutated advanced ovarian cancer, a PARP inhibitor (PARPi) can also be considered as maintenance after first-line treatment [21].

The primary challenge in the treatment of cancer is treatment resistance. Unfortunately, ovarian cancer is no exception to treatment resistance, with particular importance being the resistance to platinum compounds. Traditionally platinum resistance was defined on the basis of the duration of the response to platinum-containing chemotherapy. Patients who initially respond to platinum-based chemotherapy and relapse 6 months or longer after the initial treatment were classified as platinum sensitive, while patients who relapse within under 6 months after platinum-based chemotherapy were considered platinum resistant. Within the platinum-resistant group, a subgroup of patients presents with the worst prognosis: platinum-refractory ovarian cancer, with a disease that progresses during or within 1 month of platinum-containing first-line chemotherapy [22]. However, the current classification of platinum resistance, which is based on the 6-month platinum-free interval, has several shortcomings: the use of bevacizumab or bevacizumab with olaparib as maintenance therapy for patients responding to first-line platinum-based chemotherapy significantly prolonged PFS, rendering the evaluation of platinum response difficult [23,24]; platinum rechallenge in patients with platinum-free intervals longer than 6 months results in response rates of only 47–66% [25,26], while a platinum-free interval shorter than 6 months does not exclude a benefit from the addition of platinum to chemotherapy [27,28]. Furthermore, the studies used to define platinum resistance determined recurrence on the basis of clinical and radiological evidence of disease or clinical symptoms. These studies took place before the widespread use of CA125 to detect recurrence, and additionally, the imaging techniques available were inferior to currently used high-resolution CT, MRI, or PET-CT [22].

Most patients generally respond well to platinum-based chemotherapy, with only 20% of HGSOC presenting from the beginning with the platinum-resistant disease. However, the majority of initially platinum-sensitive patients will develop secondary platinum resistance following multiple recurrences with progressively shorter progression-free survival [15]. Therefore eventually, platinum resistance influences the prognosis of every ovarian cancer patient, representing one of the key prognostic factors influencing overall survival. In the following review, we will discuss the underlying mechanisms of platinum resistance, available biomarkers, and possibilities of overcoming resistance.

## 2. Molecular Mechanisms of Platinum Resistance in High-Grade Ovarian Cancer

Platinum compounds exert their cytotoxic anticancer effects mainly by forming covalent bonds to the DNA, thus generating DNA crosslinks and inhibiting DNA replication, eventually leading to cell death. The mechanisms of platinum resistance are multifactorial and comprise genetic and epigenetic alterations as well as immune and environmental factors frequently involving more than one mechanism of resistance [29]. Figure 1 summarizes the main mechanisms in the development of platinum resistance.

### 2.1. Alteration of Drug Influx and Efflux Pathways

One of the most-agreed-upon mechanisms of platinum resistance is the dysregulation of drug influx and efflux pathways that modulate the transport of platinum salts in the cancer cell. As a result, platinum-resistant cell lines display a reduction in cisplatin concentration, varying from 20% to 70% [30]. The copper transporter 1 (CTR-1), a transmembrane influx transporter involved in copper homeostasis, also plays a crucial role in the intracellular uptake of platinum salts. The knockout of CTR-1 in mouse cell lines led to platinum resistance via decreased intracellular platinum concentrations [31]; similarly, the overexpression of CTR-1 led to increased sensitivity to platinum in ovarian cell lines [32]. Additionally, Song et al. [33] demonstrated that the upregulation of CTR-1 expression in cisplatin-resistant small cell lung cancer cell lines restored platinum sensitivity. Ishida et al. correlated tumoral CTR-1 mRNA levels with a response to platinum-based chemotherapy in 15 patients with stage III or IV HGSOC who underwent cytoreductive surgery. Patients with platinum-sensitive disease expressed significantly higher levels of CTR-1 mRNA compared with platinum-resistant or refractory disease. These results were further validated by using clinical and array-based data from the Cancer Genome Atlas, on a subset of 91 stage III and IV HGSOC patients who underwent surgery followed by platinum-based adjuvant chemotherapy. Patients with high CTR-1 expression had significantly prolonged disease-free survival compared with those expressing low levels of CTR-1 [34]. Organic cation transporters (OCTs) are part of the solute carrier family and are involved in the cellular uptake of platinum derivates. Furthermore, low OCT6 expression has been associated with platinum resistance in human lung cancer cell lines [35].

The copper transporter 2 (CTR-2) is also involved in regulating cellular platinum levels; however, it acts as a platinum efflux transporter. Higher CTR-2 expression was linked to platinum resistance in ovarian cancer cell lines [36]. The copper exporters ATP7A and ATP7B are also involved in platinum efflux and subsequent resistance. ATP7A is responsible for the intracytoplasmic sequestration of platinum derivates blocking their access to the nucleus, while ATP7B facilitates drug efflux via the secretory pathway. The overexpression of both ATP7A and ATP7B has been associated with platinum resistance, whereas blocking their activity restores platinum sensitivity [37,38,39]. The altered expression of multidrug resistance proteins (MRPs) has been linked to multidrug resistance and worse outcomes in multiple cancers. Arts et al. [40] found that increased MRP2 and MRP4 expression was linked to platinum resistance and poor outcomes in ovarian cancer. Similarly, three other reports have associated high MRP2 levels and resistance to platinum-based chemotherapy in various cancers, including ovarian cancer [41,42,43].

### 2.2. DNA Repair

DNA is the main target of platinum-based anticancer drugs, and the cell’s ability to recognize and repair drug-induced DNA damage can influence its sensitivity or resistance to platinum chemotherapy. The primary mechanism through which platinum chemotherapy exerts its cytotoxic effects is the formation of DNA monoadducts that evolve through covalent binding to DNA crosslinks that can occur either on the same DNA strand or on the opposite strands, generating interstrand crosslinks that block DNA synthesis and transcription if they are not repaired. The DNA damage response (DDR) mechanism is activated in the presence of DNA lesions. DDR consists of several signaling pathways responsible for enforcing cell-cycle arrest and, depending on the severity of DNA damage, either DNA repair or the activation of apoptosis for cells presenting with unrepairable DNA lesions [44]. Six major DNA repair pathways have been described: mismatch repair (MMR), base excision repair (BER), nucleotide excision repair (NER), homologous recombination (HR), nonhomologous end joining (NHEJ), and Fanconi anemia (FA). An intertwined activation of these pathways is responsible for repairing DNA lesions and preventing the development of various pathologies, including cancer [45,46].

The same pathways are also accountable for preventing the accumulation of DNA lesions secondary to platinum-based chemotherapy, and their variation may promote platinum sensitivity or resistance. The upregulation of DNA repair proteins may lead to removing platinum adducts and repairing tumoral DNA, decreasing treatment efficacy. Most platinum-resistant tumors display the upregulation of DNA damage repair proteins such as BRCA 1/2, mismatch repair proteins MSH1 and MSH2, excision repair cross-complementing (ERCC) proteins, RAD51, and Fanconi anemia complementation group D2 [29,47]. BRCA1/2-mutated HGSOC have increased sensitivity to DNA-damaging agents such as PARPis and platinum agents and have an improved overall response to platinum therapy [7,48]. CDK12, a kinase involved in the HR pathway, is mutated in 3% of ovarian cancer patients. Preclinical data have associated low CDK12 expression with higher susceptibility to cisplatin and PARPis [49]. Replication protein A (RPA) recognizes single-stranded DNA lesions interfering with the replication fork and acts as an activation platform for DNA damage repair via NER. RPA-deficient ovarian cancer cells cannot efficiently repair cisplatin-induced DNA lesions via NER and display increased platinum sensitivity [50]. NER alterations are present in 8% of HGSOC and are associated with increased sensitivity to platinum chemotherapy, similar to BRCA1/2-mutated patients [51]. ERCC1, a NER-associated protein, is one of the most promising biomarkers for platinum sensitivity in these patients. Low ERCC1 expression was associated with platinum sensitivity [52,53,54], but these findings were inconsistent across multiple studies, where some reported an absent or negative correlation between ERCC1 and a response to platinum [47].

### 2.3. Epigenetic Alterations

Epigenetic processes influence gene expression without changing the DNA sequence. They are essential in ensuring normal genome functioning and ensuring altered epigenetic regulation results in the development of various pathologies, including cancer. Three key processes are involved in the epigenetic regulation of HGSOC: DNA methylation, histone modification, and microRNAs (miRs).

#### 2.3.1. DNA Methylation

DNA methylation modulates gene expression via DNA methyltransferase enzymes that catalyze the addition of a methyl group or an ethyl group onto the fifth carbon of a cytosine ring to form methylcytosine. DNA methylation frequently occurs in areas known as CpG islands, often located in the promoter region of genes. Increased cytosine methylation in the promoter region is known as hypermethylation and decreases gene expression by inhibiting transcription factors and RNA polymerase from binding DNA and undergoing transcription [55]. The role of DNA methylation in ovarian cancer chemoresistance has been extensively studied. Lum et al. [56] analyzed DNA methylation in 36 HGSOC samples segregated on the basis of platinum sensitivity. They identified 749 probes corresponding to 296 genes that were significantly differently methylated in platinum-sensitive samples and in platinum-resistant samples; furthermore, they observed that hypermethylation was more often present in platinum-resistant samples than in platinum-sensitive ones. Two other reports found the same association between hypermethylation and platinum resistance [57,58]; however, these findings are inconsistent across studies. Lund et al. [59] found that the majority (1251 of 1488) of the differentially methylated sites were hypomethylated in cisplatin-resistant samples. A pathway analysis of the 452 hypermethylated genes associated with platinum resistance, by Cardenas et al. [58], found the epithelial–mesenchymal transition (EMT) pathway to be the most influenced by aberrant methylation in the development of the chemoresistant phenotype. MSX1 encodes a member of the muscle segment homeobox gene family and can influence EMT in ovarian cancer. The hypomethylation of MSX1 leads to decreased MSX1 expression, which is associated with cisplatin resistance in ovarian cancer cell lines, while MSX1 overexpression sensitizes cells to cisplatin [60]. LAMA3 (laminin alpha 3), a component of the cell base membrane, plays an important role in cell adhesion, migration, and embryo differentiation. Reduced LAMA3 expression has been associated with EMT in various tumors, including ovarian cancer. Feng et al. demonstrated that the hypermethylation of LAMA3 was responsible for the reduced expression and that decreased LAMA3 levels were correlated with chemoresistance and poor outcomes [61]. The SOX9, ZIC1, and TWIST genes involved in EMT were also associated with a hypermethylated status in platinum-resistant ovarian cancer [56]. The aberrant methylation of genes involved in the wingless/integrated (Wnt) signaling pathway was also associated with platinum resistance in HGSOC. FZD1, FZD10, and GSK3B were found to be differentially methylated in both platinum-resistant samples and platinum-sensitive ones [62]. Wang et al. found that DNA methylation via the PI3K-Akt pathway is associated with low BRCA1 expression in ovarian cancer cell lines, and BRCA1 demethylation was associated with the development of platinum resistance [63].

#### 2.3.2. Histone Modifications

Histone modifications, regulated by histone-modifying enzymes, directly affect gene expression by altering the chromatin structure. Histones are susceptible to several changes, including acetylation, methylation, phosphorylation, ubiquitination, glycosylation, sumoylation, ADP-ribosylation, and carbonylation. However, histone acetylation is of particular importance as it has been associated with ovarian cancer pathogenesis [64]. Histone acetyltransferase (HAT) enzymes add acetyl groups to the histone surface, enabling RNA polymerase II interaction and favoring gene expression. Meanwhile, histone deacetylase (HDAC) enzymes remove acetyl groups from histones and increase chromatin compaction, thus restricting RNA polymerase II access with subsequently decreased gene expression [62]. Cacan et al. [65] demonstrated HDAC1 involvement in cisplatin resistance in ovarian cancer cells. The suppression of HDAC1 and DNA methyltransferase activity in platinum-resistant ovarian cancer cells restored cisplatin-mediated cell deaths through the upregulation of RGS10, an essential regulator of cell survival and chemoresistance. Liu et al. [66] demonstrated that HDAC1 knockdown in cisplatin-resistant cell lines suppressed proliferation and increased apoptosis and chemosensitivity through the downregulation of the c-Myc oncogene and the upregulation of miR-34a. Furthermore, cisplatin treatment in platinum-sensitive cells increased HDAC1 and c-Myc expression while inactivating miR-34a, leading cells to acquire chemoresistance to cisplatin.

#### 2.3.3. MicroRNAs

MicroRNAs are small 19–25-nucleotides-long single-stranded noncoding RNAs in the post-translational regulation of gene expression. Multiple miRs have altered expression in HGSOC and are associated with carcinogenesis, progression, metastasis, and drug resistance [67]. MiR-mediated platinum resistance arises through multiple mechanisms influenced by microRNA dysregulation. MiR-130a was found to be involved in platinum resistance occurrence by altering cellular cisplatin uptake. The overexpression of miR-130a was associated with platinum resistance by targeting the SOX9/miR-130a/CTR1 axis [68]. Cisplatin resistance can also be secondary to increased cellular drug efflux. The ATP7A and ATP7B transporters are associated with platinum chemotherapy resistance and are influenced by miRs expression. MiR-139 dysregulation influences ATP7A and ATP7B expression with secondary platinum resistance. Platinum-resistant cell lines presented low levels of MiR-139 and high ATP7A and ATP7B expression. MiR-139 overexpression enhanced the suppressive effect of cisplatin on resistant cell lines. Furthermore, there is an inverse correlation between miR-139 and ATP7A/B expression [69]. MiR-15a and miR-16 are also involved in ATP7B regulation and platinum resistance. MiR-15a and miR-16 transfection in cisplatin-resistant cell lines and murine models have restored cisplatin sensitivity by inhibiting ATP7B expression [70]. MiR also influences MRP2-associated resistance. The upregulation of miR-490-3p and downregulation of miR-411 was associated with increased cisplatin sensitivity via the inhibition of MRP2 expression ovarian cell lines [71,72]. MiR-514 downregulation was associated with advanced stages of and poor outcomes in ovarian cancer. MiR-514 also increases cisplatin chemosensitivity by targeting ATP-binding cassette subfamily members ABCA1, ABCA10, and ABCF2 [73].

MicroRNA modulation influences pathways involved in the process of DNA repair and the secondary platinum resistance induced by their activation. One study demonstrated that miR-211 expression enhanced platinum sensitivity in ovarian cancer cells by targeting DDR. MiR-211 facilitated platinum-induced DNA damage by targeting DDR effector genes, including POLH, TDP1, ATRX, MRPS11, and ERCC6L2 [74]. Enhanced nucleotide excision repair is associated with resistance to platinum chemotherapy. ERCC1, an essential effector of the NER pathway, is characterized as a potential biomarker for platinum resistance and is a direct target of miR-30a-3p. Increased miR-30a-3p restored cisplatin sensitivity by targeting ERCC1 and ATP7A [75]. NER-pathway-induced resistance is also influenced by miR-770-5p expression. Downregulated in cisplatin-resistant cell lines, miR-770-5p overexpression restored cisplatin sensitivity by directly targeting ERCC2, an effector of the NER pathway [76]. Zhu et al. [77] also demonstrated miR-770-5p involvement in cisplatin resistance; the long noncoding RNA nuclear paraspeckle assembly transcript 1 (NEAT1) has been shown to enable treatment resistance by inhibiting miR-770-5p and upregulating PARP1 expression, a promoter of platinum resistance. MiR-9 inhibits homologous recombination-associated resistance by targeting BRCA1. Patients with high MiR-9 expression have better chemotherapy responses and increased platinum sensitivity. MiR-9 levels were inversely correlated with BRCA1 expression and treatment with miR-9-sensitized BRCA1-proficient cell lines to cisplatin [78]. MiR-506 and miR-152 can increase platinum sensitivity by targeting RAD51 and suppressing HR [79,80]. Choi et al. demonstrated that miR-622 could be responsible for platinum and PARPi resistance in BRCA1-mutated tumors by restoring HR-mediated double-stranded break repair [81]. MiR-146a, miR-148a, and miR-545 are linked to improved outcomes in ovarian cancer patients by targeting BRCA1/2 expression [82]. In contrast, miR-493-5p expression promotes platinum and PARPi resistance in BRCA2-mutated ovarian carcinoma by reducing nucleases and other factors involved in maintaining genomic stability, thus resulting in relatively stable replication forks, diminished single-strand annealing, and increased R-loop formation [83]. The epigenetic mechanisms of resistance are also influenced by microRNA modulation. Liu et al. demonstrated that the upregulation of miR-200b and miR-200c restored cisplatin cytotoxicity by directly targeting the DNA methyltransferases (DNMT) responsible for DNA methylation, often associated with treatment resistance [84]. Low levels of miR-30a-5p and miR-30c-5p are associated with cisplatin resistance through DNMT upregulation and subsequent hypermethylation. DNMT1 is a direct target of miR-30a-5p and miR-30c-5p, and the overexpression of miR-30a-5p and miR-30c-5p-inhibited DNMT1 promoted cisplatin sensitivity and partially reversed EMT in ovarian cancer cell lines [85]. MiR-152 and miR-185 were also found to be downregulated in platinum-resistant ovarian cell lines, and their upregulation reversed cisplatin sensitivity, increased apoptosis, and inhibited proliferation by targeting DNMT1 [86].

Robust data suggest an EMT association with platinum resistance in ovarian cancer. MicroRNAs mediate platinum resistance or sensitivity by regulating EMT [87]. MiR-186 downregulation was associated with EMT and chemoresistance by targeting Twist1 in ovarian cancer cell lines [88]. MiR-363 low expression was also linked to chemoresistance and carcinogenesis via Snail-induced EMT [89]. Zhan et al. demonstrated that miR-1294 is downregulated in cisplatin-resistant ovarian cancer cell lines and that the overexpression of miR-1294 prevented platinum resistance by directly targeting IGF1R and inhibiting EMT [90]. MiR-20a promotes a cisplatin-resistant phenotype in ovarian cancer cells by activating EMT [91]. High oncogenic miR-205-5p and miR-216a levels were linked to platinum resistance in ovarian cancer cell lines by targeting the PTEN/Akt pathway [92,93]. MiR-483-3p and miR-224-5p conferred platinum resistance by suppressing protein kinase C family members [94,95,96]. MiR-1180 was associated with bone-marrow-derived mesenchymal stem-cell-induced platinum resistance in HGSOC cells. Mir-1180 overexpression leads to Wnt signaling and secondary glycolysis-induced chemoresistance [97]. A high expression of the platinum-refractory phenotype miR-98-5p directly targets Dicer1 and suppresses its activity, causing global miR downregulation; additionally, it inhibits cyclin-dependent kinase inhibitor 1A (CDKN1A), a promoter of cisplatin sensitivity [80,98]. Table 1 summarizes microRNA involvement in ovarian cancer platinum resistance.

### 2.4. Tumoral Microenvironment

Ovarian cancer arises in a unique tumoral microenvironment (TME) that plays a crucial part in the natural history of the disease. The TME comprises stromal cells, immune cells, endothelial cells, adipocytes, bone-marrow-derived cells, lymphocytes, and the extracellular matrix (ECM), which play essential roles in supporting tumor progression through signaling molecules that promote cell growth, differentiation, and invasiveness. Unlike the cells of other epithelial tumors, ovarian cancer cells detach from their origin in the ovary and the fallopian tube and adhere to the mesothelial layers of the peritoneum, covering the abdominal organs and invading the submesothelial layers. In addition, ovarian cancer cells can survive in the ascitic fluid, which acts as a medium wherein tumor cells disseminate throughout the entire abdominal cavity. Alongside ovarian tumor cells, the ascitic fluid also includes multiple types of nontumorigenic cells regulated by soluble factors and extracellular vesicles that promote tumor growth and metastasis [99,100,101,102].

The extracellular matrix consists of glycosaminoglycans, proteoglycans, hyaluronan, collagen, fibronectin, vitronectin, elastin, laminin, and other glycoproteins that sustain tissue integrity but also regulate cell migration, growth, and protein synthesis [102,103]. In ovarian cancer, the ECM signaling is dysregulated through the activation of cancer-associated fibroblasts (CAFs) and tumor-associated macrophages (TAMs), which leads to excessive ECM remodeling associated with tumor progression but also treatment resistance through the activation of multiple signaling pathways [99]. Osterman et al. demonstrated the role of ECM in promoting platinum resistance in ovarian cancer, in which ECM inhibits focal adhesion kinase (FAK), a cytosolic tyrosine kinase activated by matrix and integrin receptors that controls cell motility. High FAK expression is associated with ovarian cancer cells resistant to platinum chemotherapy. Combining FAK inhibition with platinum chemotherapy overcame this resistance and increased apoptosis [104]. Cell-adhesion-mediated drug resistance (CAM-DR) enables cells to rapidly evade cytotoxic stress by interacting with the elements of the ECM. CAM-DR markers CD44, basigin (CD147), HE4, integrin α5, and β1 were elevated in chemoresistant HGSOC patients and were associated with poor outcomes [105]. Growing ovarian cancer cells in collagen type 1 decreased their platinum sensitivity by activating CAM-DR via integrin β1. Integrin β1 knockdown restored platinum sensitivity in platinum-sensitive ovarian cell lines but not in platinum-resistant ones, suggesting CAM-DR activation via integrin β1 as an initial mechanism of resistance in ovarian cancer [106]. Proteomic profiling of chemoresistant HGSOC revealed the overexpression of 10 ECM-associated proteins, specifically decorin, versican, CD147, fibulin-1, extracellular matrix protein 1, biglycan, fibronectin 1, dermatopontin, alpha-cardiac actin, and an EGF-containing fibulin-like extracellular matrix protein 1 [107]. Additionally, carboplatin treatment increased hyaluronan expression in ovarian cancer cells, leading to chemoresistance by the upregulation of the membrane ATP-binding cassette transporter proteins (ABCB3, ABCC1, ABCC2, and ABCC3) in CD44-expressing ovarian cells. Treatment with hyaluronan oligomers restored platinum sensitivity in chemoresistant cells [108].

Ovarian cancer cell and mesothelial cell crosstalk promotes tumor adhesion and invasion, but ovarian-cancer-associated mesothelial cells also induce chemoresistance through the ATP-binding cassette transporter protein induction of the fibronectin 1/Akt signaling pathway [109]. Cancer-associated fibroblasts (CAFs) occur in the TME secondary to inflammation and hypoxia. They promote tumor growth, proliferation, and metastasis; inhibit immune regulation; and modulate cell metabolism but are also involved in treatment resistance [110]. CAFs can obstruct chemotherapy transport to the cancer cell by creating physical barriers and microvascular compression. Additionally, they can mediate resistance by secreting cysteine and glutathione, thus reducing the intracellular concentration of cisplatin via competition to DNA binding sites and platinum efflux through an ATP-dependent glutathione S-conjugate export pump [111]. Functional studies have revealed that CAFs and cancer-associated adipocytes (CAAs) are also able to transfer miR-21 to the ovarian cancer cell, where it inhibits apoptosis and confers chemoresistance by the downregulation of APAF1 [112]. CAAs represent essential elements of the ovarian cancer milieu, promoting metastasis and chemoresistance. Lipidomic analysis found that CAAs were responsible for the secretion of arachidonic acid, a chemoprotective lipid mediator that acts directly on the ovarian tumor cell and inhibits cisplatin-induced apoptosis through Akt pathway activation [113]. Tumor-associated macrophages (TAMs) were also found to promote chemoresistance. Hypoxic TAMs were responsible for the exosomal transfer of miR-223 to the ovarian cancer cells that promote drug resistance by activating the PTEN-PI3K/AKT pathway [114].

## 3. Overcoming Platinum Resistance in Ovarian Cancer

Platinum resistance is one of the most important prognostic factors in ovarian cancer and one of the main factors driving HGSOC mortality. Therefore, overcoming platinum resistance is considered one of the most significant challenges in ovarian cancer. The current management of platinum-resistant disease involves treatment with nonplatinum chemotherapy, such as paclitaxel, pegylated liposomal doxorubicin, or topotecan alone or in association with the antiangiogenic agent bevacizumab, which improved PFS compared to chemotherapy alone [115]. Alternative treatment strategies may include gemcitabine or etoposide. Nevertheless, platinum rechallenge can also be an option even for platinum-resistant disease. Various studies demonstrated longer PFS and higher response rates for platinum-based associations compared with monotherapy, especially in patients with a platinum-free interval longer than 3 months. However, new biomarkers that may enable the selection of patients that benefit from this strategy are needed [27,116,117,118].

PARP inhibitors make up a class of drugs that inhibits the activity of an alternate DNA repair pathway. Single-strand DNA breaks are detected by the PARP family of proteins that initiate DNA repair through the BER pathway. PARPis block the activity of PARP1, leading to the accumulation of single-strand DNA breaks and, eventually, double-stranded DNA breaks, which only a functional HR pathway can repair. Therefore, PARPis exploit HR deficiency to promote cancer cell death [119]. Although platinum and PAPRis share a common mechanism of resistance, specifically through the reactivation of the HR pathway, PARPis are an option worth exploring in the management of platinum-resistant disease. Kaufman et al. [120] reported an objective response rate of 31.1% and stable disease in 40.4% of the platinum-resistant BRCA-mutated ovarian cancer patients treated with olaparib. A similar response rate of 33.5% was reported by Fong et al. They demonstrated a clear connection between the platinum response and the clinical benefit of olaparib in BRCA-mutated ovarian cancer. Further, 61.5% of the platinum-sensitive patients responded (partial or complete response) according to the RECIS or GCIG criteria, compared with 41.7% in the platinum-resistant group. Platinum-refractory patients had the lowest response rates; there were no radiologic responders, and only one patient had stable disease lasting for more than four cycles [121]. Similar response rates were reported for rucaparib, niraparib, and veliparib administration in the setting of platinum-resistant HGSOC [121,122,123,124].

Recently, combinational therapy with PARPi has gained attention. The association between PARPis and antiangiogenic agents was investigated across several clinical trials. Niraparib and the antiangiogenic tyrosine kinase inhibitor (TKI) anlotinib demonstrated promising objective response rates (ORR): 50% with a PFS of 9.2 months in platinum-resistant ovarian cancer patients [125]. A combined treatment of olaparib and bevacizumab resulted in a superior response and 3-year survival compared with bevacizumab and albumin-bound paclitaxel [126]. However, the association of cediranib and olaparib failed to achieve superior outcomes in platinum-resistant disease compared with chemotherapy [127,128].

Ataxia telangiectasia and RAD3-related protein kinase (ATR)/checkpoint kinase 1 (CHK1) have attracted significant attention as possible targets for anticancer therapy because of their role in regulating cell-cycle checkpoints. The ATR/CHK1 pathway acts as a sensor detecting single-stranded DNA breaks that lead to cell-cycle arrest. Combined ATR and PARP inhibition has been evaluated across multiple studies; despite promising preclinical data, the phase 2 CAPIRI trial failed to demonstrate a clinical benefit in platinum-resistant epithelial ovarian cancer [129,130]. Prexasertib, a CHK1 inhibitor, was also evaluated in BRCA wild-type HGSOC, where the majority (79%) of the patients had platinum-resistant or refractory disease. Prexasertib showed clinical activity, where 33% of the patients had a partial response (PR) and 29% had stable disease (SD) [131]. WEE-1 inhibitors target the WEE-1 kinase, a G2 cell-cycle checkpoint regulator, resulting in increased apoptosis secondary to the accumulation of irreparable genetic lesions [132]. The WEE-1 inhibitor AZD1775 was evaluated in a phase 2 trial and demonstrated clinical activity, with a 43% ORR and 5.3-month PFS in p53-mutated platinum-resistant or refractory ovarian cancer patients [132]. BET inhibitors bind the bromodomains of BET proteins interfering with BRCA1 and RAD51 expression. BET inhibition in ovarian cancer cell lines resulted in HR deficiency, thus providing an argument for combined BET and PARP inhibition. Olaparib combined with various BET acted synergistically, increasing either treatment’s efficacy alone, irrespective of HR status. Additionally, the association of BET inhibition with cisplatin chemotherapy exhibited the same synergistic activity. The coadministration of cisplatin and BET inhibitors increased ovarian cancer cells’ sensitivity to cisplatin even in resistant cell lines [133,134].

Epigenetic dysregulation is involved in the acquisition of the platinum-resistant phenotype through multiple mechanisms; thus, epigenetic modulators have been investigated as potential therapies to reverse platinum resistance and resensitize tumors to platinum salts. DNMT inhibitors showed modest clinical activity in monotherapy, but combined treatment may enhance sensitivity to platinum compounds. When combined with carboplatin, the DNMT inhibitor guadecitabine showed a superior 6-month PFS compared with physicians’ choice treatment: 37% vs. 11% [135]. Similarly, combining carboplatin with low-dose decitabine resulted in a clinical benefit rate of 70%, with an ORR of 35% and a median PFS of 309 days [136]. Hypermethylation has been associated with an immunosuppressive tumoral milieu by silencing tumoral antigen expression and downregulating programmed death ligand (PDL) expression [137,138]. On the basis of these findings, it was hypothesized that the association of epigenetic therapy and immune checkpoint inhibitors (ICIs) could boost ovarian cancer tumoral immunogenicity and increase ICI efficiency [139]. Chen et al. [140] evaluated the hypomethylating agent guadecitabine in association with pembrolizumab in 35 platinum-resistant ovarian cancer patients, where 8.6% of the patients had PR and 22.9% SD, resulting in a clinical benefit rate of 31.4%, with a median response duration of 6.8 months. The association of the CC-486 hypomethylating agent and durvalumab was also investigated in a phase II basket trial that included platinum-resistant ovarian cancer; however, the association failed to achieve any clinical activity [141].

HDAC inhibitors were also evaluated; however, they failed to demonstrate consistent efficacy across studies [142,143]. An association between avelumab and entinostat, a class I selective HDAC inhibitor, was also assessed in pretreated ovarian cancer patients but failed to improve PFS compared with avelumab alone [144]. The association between HDAC inhibitors and DNMT inhibitors was also evaluated to determine their synergistic activity [145]. Falchook et al. [146] investigated the association of azacytidine and valproic acid in restoring carboplatin sensitivity in a phase 1 trial, with a clinical benefit rate of 18.8% but with high toxicity, where 81% of the patients reported grade ≥ 3 adverse events, including fatigue, neutropenia, and vomiting. Preclinical models evaluated the association between immunotherapy and the combination of DNMT1 with an enhancer of zeste homologue 2 (EZH2) inhibition in ovarian cancer cells. EZH2-mediated histone H3 lysine 27 trimethylation and DNMT1-mediated DNA methylation were shown to repress the production of the T helper 1 type of chemokines: CXC-motif chemokine 9 (CXCL9) and CXCL10. Combined EZH2 and DNMT1 inhibition increased effector T-cell tumor infiltration, inhibited tumor progression, and improved the therapeutic efficacy of PDL-1 blockade [147].

An immune checkpoint blockade aims to restore T-cell function and reverse tumor-associated immune-evasion mechanisms, with the aim of producing a sustained T-cell-mediated antitumoral response. Unfortunately, despite promising results in various solid tumors, checkpoint inhibition has failed to provide a significant benefit in ovarian cancer. Immune checkpoint inhibitor monotherapy with nivolumab, pembrolizumab, avelumab, or atezolizumab showed a favorable toxicity profile but was unable to provide substantial clinical benefit, with an ORR of 6–22% [148].

The disappointing efficacy of ICI monotherapy represented the rationale for investigating ICI-combined treatment strategies. One promising combination is ICIs and PARPi because HR-deficient tumors display high PD-1 expression, and the accumulation of double-stranded DNA breaks enables the buildup of neoantigens [149]. The efficacy of the niraparib and pembrolizumab combination was assessed in recurrent platinum-resistant ovarian cancer patients. An ORR of 18% with a disease control rate of 65% was observed irrespective of platinum sensitivity, BRCA, or HR status [150]. Lampert et al. evaluated a durvalumab and olaparib combination in recurrent ovarian cancer. Most patients were platinum resistant (86%) and had the BRCA wild type (77%). Although the disease control rate was 71%, the clinical activity was modest, with an ORR of 14% [151].

The association between antiangiogenic therapy and ICI was also investigated. Hypoxia and VEGF dysregulation promote an immunosuppressive microenvironment by shifting the T helper 1 antitumoral response to a T helper 2 protumorigenic response; antigen presentation by dendritic cells is also inhibited; and VEGF itself has immunosuppressive properties [152]. Liu et al. [153] assessed the efficacy of a nivolumab and bevacizumab combination in relapsed ovarian cancer. The ORR was 40% in platinum-sensitive and 16.7% in platinum-resistant disease. The median PFS was 7.7 months in the platinum-resistant subgroup and 12.1 months in the platinum-sensitive one. Bevacizumab was also evaluated in combination with pembrolizumab and cyclophosphamide in recurrent ovarian cancer. Patients with platinum-resistant disease had an ORR of 43.3%, where 93.3% of patients exhibited a clinical benefit and had a 5.5-month median duration of response [154].

Chemotherapy was shown to induce an immunogenic antitumoral response and promote a proinflammatory tumoral microenvironment through the release of inflammatory signals from dying tumor cells [155]. This rationale was the basis for investigating the safety and efficacy of chemotherapy and ICI association in multiple solid tumors, including ovarian cancer. The JAVELIN Ovarian 200 trial evaluated compared avelumab and pegylated liposomal doxorubicin (PLD) monotherapy to the avelumab and PLD combination in 566 platinum-resistant ovarian cancer patients. Neither avelumab monotherapy nor the combination of avelumab and PLD improved PFS or OS compared with PLD monotherapy. However, there was a higher ORR for the combo in the PDL1-positive group compared with the PDL1-negative one: 18.5% vs. 3.4%. This ORR also translated into a survival advantage for the PDL1-positive patient subgroup [156]. The association between PLD and pembrolizumab was also evaluated in 23 platinum-resistant ovarian cancer patients, where 52.2% of patients achieved a clinical benefit from the combinational treatment, with an ORR of 26.1% and a favorable toxicity profile. There was no significant correlation between PDL1 expression and an objective response [157].

Copper transporter dysregulation has been validated as a critical mechanism of platinum resistance in HGSOC, and this is the basis for targeting copper homeostasis as a mechanism to resensitize ovarian cancer cells to platinum compounds. Using cisplatin-resistant ovarian cancer cell lines, Liang et al. [158] demonstrated that cisplatin resistance is associated with the decreased expression of the high-affinity copper transporter 1 (hCTR1). Furthermore, they revealed that copper chelators resensitize cells to cisplatin by enhancing hCTR1 expression. Following this preclinical data, the association between carboplatin and the copper-lowering agent trientine was evaluated in platinum-resistant patients. The association was well tolerated and displayed antitumor activity, especially in patients with lowered ceruloplasmin and copper levels, but the response rates remained low, warranting improvement [159]. The association between trientine carboplatin and PLD was also assessed in a dose escalation study involving patients with relapsed epithelial ovarian, tubal, and peritoneal cancers. The combination was well tolerated and safe, rendering a clinical benefit rate of 33.3% in the platinum-resistant group and 50% in the partially platinum-sensitive group [160]. Tranilast (an analog of tryptophan metabolite and an inhibitor of histamine release) and telmisartan (an angiotensin II receptor antagonist) were shown to facilitate platinum compound delivery to the nucleus by targeting ATP7B expression and trafficking in platinum-resistant IGROV-CP20 ovarian cancer cell lines. Amphotericin B was also shown to promote cisplatin toxicity by inhibiting ATP7B expression but with an inferior safety profile compared with tranilast and telmisartan [161]. Theaflavin-3,3′-digallate (TF3), a black tea polyphenol, was also shown to enhance ovarian cancer cells’ sensitivity to cisplatin. TF3 increased the intracellular accumulation of cisplatin and enhanced platinum DNA damage by decreasing glutathione levels and upregulating CTR1 levels [162].

## 4. Conclusions

Alongside surgery, chemotherapy is the cornerstone of treatment in advanced ovarian cancer, and platinum-based combinations continue to be the most effective first-line treatment for these patients. Despite the initial efficacy, most patients will present recurrent disease. Rechallenge with platinum-based chemotherapy is the treatment of choice for patients with platinum-sensitive disease, defined as a platinum-free interval longer than 6 months. These patients usually respond to platinum rechallenge and have a better prognosis than those who are platinum resistant. Therefore, we can safely consider platinum resistance as one of the most important prognostic factors in ovarian cancer.

Although classically defined on the basis of the 6-month platinum-free cutoff interval, the concept of platinum resistance is an everchanging concept owing to the widespread availability of CA125, high-resolution and functional imaging that enables early recurrence detection, and the changes in maintenance therapy now that bevacizumab and PARPis have managed to prolong PFS, thus delaying recurrence. Nevertheless, resistance to platinum-based cytotoxic agents is a complex concept resulting from an interplay between mechanisms. Tumor cells can modify the intracellular concentration of chemotherapy by changing the expression of cellular influx and efflux transporters. Changes in the DNA repair pathways are involved in platinum resistance, but they can also represent targetable therapeutic opportunities. Recent data have revealed that the dysregulation of epigenetic control processes with aberrant miR expression, histone acetylation, and DNA methylation modulates these resistance mechanisms’ expression. The tumoral microenvironment is essential in ovarian cancer carcinogenesis and progression but can also promote treatment resistance through EMT and various adaptative signals modulated by the stromal cells of the tumoral milieu. Despite the substantial progress in understanding the underlying mechanisms of platinum resistance, more research is necessary to understand their interplay and contribution to achieving the resistant phenotype.

Several treatment strategies have been evaluated in the setting of platinum-resistant ovarian cancer, and treatment associations involving PARP inhibition, antiangiogenic agents, immune checkpoint inhibitors, and chemotherapy have shown promising results. However, more research is necessary to identify biomarkers that enable better patient stratification.

## Figures and Tables

**Figure 1 medicina-59-00544-f001:**
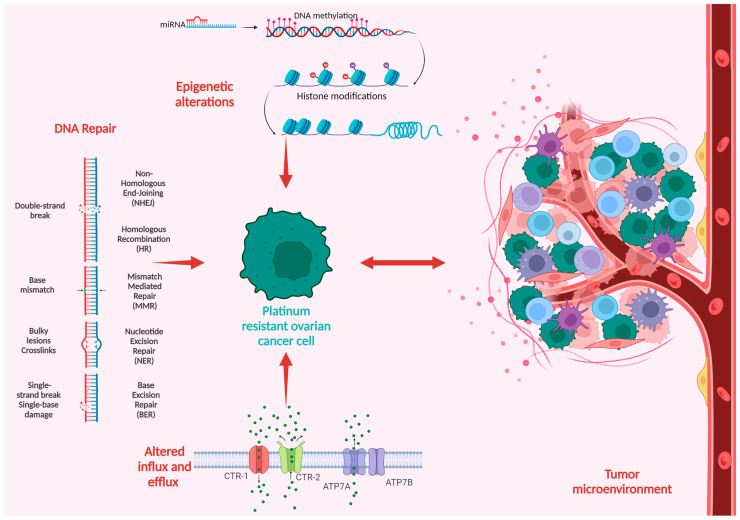
Schematic overview of the mechanisms of platinum resistance in ovarian cancer.

**Table 1 medicina-59-00544-t001:** MicroRNA involvement in HGSOC platinum resistance.

MicroRNA	Target Gene	Effect on Cisplatin Response	Ref.
miR-130a	SOX9/miR-130a/CTR1 axis	Resistance	[68]
miR-411	MRP2		[72]
miR-622	Ku70, Ku80		[81]
miR-20a	EMT		[91]
miR-205-5p	PTEN		[92]
miR-216a	STAT3/miR-216a/PTEN axis		[93]
miRr-483-3p	PKC-alpha		[95]
miR-224-5p	PKC-delta		[96]
mir-1180	Wnt		[97]
miR-98-5p	Dicer1, CDKN1A		[98]
miR-493-5p	MRE11, CHD4, EXO1, RNASEH2A, FEN1, SSRP1		[83]
miR-139	ATP7A/B	Sensitivity	[69]
miR-15amiR-16	ATP7B		[70]
miR-490-3p	MRP2		[71]
miR-514	ABCA1, ABCA10, ABCF2		[73]
miR-211	POLH, TDP1, ATRX, MRPS11, ERCC6L2		[74]
miR-30a-3p	ERCC1		[75]
miR-770-5p	ERCC2, NEAT1		[76,77]
miR-9	BRCA1		[78]
miR-506	RAD51		[79]
miR-152	RAD51, DNMT		[80,86]
miR-146a, miR-148a, miR-545	BRCA1/2		[83]
miR-200b, miR-200c	DNMT		[84]
miR-30a-5p, miR-30c-5p	DNMT		[85]
miR-185	DNMT		[86]
miR-186	Twist1		[88]
miR-363	Snail-induced EMT		[89]
miR-1294	IGF1R		[90]

## Data Availability

Data sharing not applicable.

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
