# Peer review of "Ovarian Cancer—Insights into Platinum Resistance and Overcoming It"

_medicina, 2023, doi:10.3390/medicina59030544_

Round 1

Reviewer 1 Report

“Ovarian cancer – insights into platinum resistance and overcoming it” by Havasi et al outlines the molecular mechanisms at play in platinum resistant HGSOC including alteration of drug pathways, upregulation of DNA repair, epigenetic alternations and the tumor microenvironment. The introductory section expertly discusses the mutational status and the standard of treatment for both platinum-sensitive and platinum resistant high-grade serous ovarian cancers (HGSOCs). Each potential mechanism of platinum-resistance is well-defined with appropriate references and Figure 1 is an excellent overall summary of the mechanisms discussed throughout the review.

Platinum-resistant HGSOC patients are an obviously highly unmet population with limited therapeutic options and the authors admirably depicted the unmet therapeutic need. The discussion of each of the four postulated resistant mechanisms was also well done. The section about overcoming platinum resistance does, however, leave some to be desired. The therapeutic conversation includes an overemphasis on treatment options for BRCA-mutated patients who, as the authors acknowledge in line 42, account for less than 15% of HGOSC patients. It would benefit the overall completeness of the review to include therapeutic options related to all discussed resistance mechanisms. While not as common as other therapeutic options under investigation, enhancement of copper transporters to re-sensitize platinum-resistance cancers has been evaluated. Discussion of some the preclinical work as well as clinical trials looking at trientine (such as NCT01178112, NCT03480750) should be added. The potential therapies to overcome epigenetic dysregulation should be expanded upon as well.

Author Response

We want to thank the reviewer for reviewing our manuscript. We are confident that we will improve the current paper by answering the reviewer’s requests. 

We have amended our manuscript according to the reviewer’s suggestions. The changes were done using track changes.

We have improved our section on overcoming platinum resistance, including data regarding the modulation of copper transporter expression in overcoming resistance.

Thank you for taking the time and analyzing our manuscript.

Reviewer 2 Report

This manuscript reviews data on the molecular basis for platinum-resistant ovarian cancer and discusses various treatment strategies.  The review is timely, comprehensive and content-rich. It is also very well-written. The manuscript would be improved if the following points were addressed.

1.     The font size of some of the text in Figure 1 is small and difficult to read.

2.     The authors provide a comprehensive discussion of the microRNA data. A table summarizing this data, perhaps listing the microRNA, the target and the functional effects would be very helpful to the reader.

3.     In the sentence beginning on line 83, one copy of the word “maintenance” can be removed.

4.     On line 363, ECM rather than EMC is meant.

5.     On line 385, please write out fibronectin. The FN1 abbreviation is not used elsewhere.

Author Response

We want to thank the reviewer for reviewing our manuscript. We are confident that we will improve the current paper by answering the reviewer’s requests. 

We have amended our manuscript according to the reviewer’s suggestions. The changes were done using track changes.

1. We modified figure 1 according to your suggestions; the font size and transparency have been changed 

2. A new table was introduced summarizing the miRNA data

3-5. Changes were made according to your suggestions.

Thank you for taking the time and analyzing our manuscript.